



Atmospheric
Chemistry
and Physics

# On the impact of future climate change on tropopause folds and tropospheric ozone

Dimitris Akritidis[1], Andrea Pozzer[2], and Prodromos Zanis[1]

[1]Department of Meteorology and Climatology, School of Geology, Aristotle University of Thessaloniki, Thessaloniki, Greece TS1
[2]Max Planck Institute for Chemistry, Mainz, Germany

**Correspondence:** Dimitris Akritidis (dakritid@geo.auth.gr)

**Abstract.** TS2 CE1 Using a transient simulation for the period 1960–2100 with the state-of-the-art ECHAM5/MESSy Atmospheric Chemistry (EMAC) global model and a tropopause fold identification algorithm, we explore the future projected changes in tropopause folds, stratosphere-to-troposphere transport (STT) of ozone, and tropospheric ozone under the RCP6.0 scenario. Statistically significant changes in tropopause fold frequencies from 1970–1999 to 2070–2099 are identified in both hemispheres, regionally exceeding 3 %, and are associated with the projected changes in the position and intensity of the subtropical jet streams. A strengthening of ozone STT is projected for the future in both hemispheres, with an induced increase in transported stratospheric ozone tracer throughout the whole troposphere, reaching up to $10 \, \text{nmol mol}^{-1}$ TS3 in the upper troposphere, $8 \, \text{nmol mol}^{-1}$ in the middle troposphere, and $3 \, \text{nmol mol}^{-1}$ near the surface. Notably, the regions exhibiting the largest changes of ozone STT at 400 hPa coincide with those with the highest fold frequency changes, highlighting the role of the tropopause folding mechanism in STT processes under a changing climate. For both the eastern Mediterranean and Middle East (EMME) and Afghanistan (AFG) regions, which are known as hotspots of fold activity and ozone STT during the summer period, the year-to-year variability of middle-tropospheric ozone with stratospheric origin is largely explained by the short-term variations in ozone at 150 hPa and tropopause fold frequency. Finally, ozone in the lower troposphere is projected to decrease under the RCP6.0 scenario during MAM (March, April, and May) and JJA (June, July, and August) in the Northern Hemisphere and during DJF (December, January, and February) in the Southern Hemisphere, due to the decline of ozone precursor emissions and the enhanced ozone loss from higher water vapour abundances, while in the rest of the troposphere ozone shows a remarkable increase owing mainly to the STT strengthening and the stratospheric ozone recovery.

## 1 Introduction

Tropospheric ozone plays a key role in the oxidizing capacity of the atmosphere (Lelieveld et al., 2016); it is also a short-lived climate forcer, being an important greenhouse gas, while near the surface it is a pollutant detrimental to human health, crops, and ecosystems (Monks et al., 2015). The future tropospheric ozone changes on a global scale depend on changes of the processes that control tropospheric ozone budget, namely chemical ozone production and loss, stratosphere–troposphere exchange (STE), and deposition (Young et al., 2013). The net stratospheric influx results from STE processes, comprised of troposphere-to-stratosphere transport (TST) and stratosphere-to-troposphere transport (STT) with tropopause folds considered to be the main mechanism for stratospheric intrusions in STT events (Stohl et al., 2003). In the 21st century, emissions of ozone precursor species, ozone-depleting substances (ODSs), and long-lived greenhouse gases (GHGs) are expected to be the major factors governing ozone amounts and its distribution in the troposphere and the stratosphere (Fiore et al., 2015; Revell et al., 2015). More specifically, future changes of

the net stratospheric influx in STE are linked to changes of the stratospheric Brewer–Dobson circulation (BDC) and the amount of ozone in the lowermost stratosphere, which are strongly influenced in a changing climate by the emissions of ODSs and GHGs (Oberländer-Hayn et al., 2016; Morgenstern et al., 2018). Moreover, climate-related changes in lightning $NO_x$ emissions, biogenic volatile organic compound (BVOC) emissions, and water vapour content are also key drivers of future tropospheric ozone changes, affecting its chemical production and loss processes (Wild, 2007; Fiore et al., 2012, 2015; Doherty et al., 2013).

Predominantly, the foldings of the tropopause are of limited vertical extent and their global spatio-temporal distribution is mainly controlled by the location and intensity of the jet stream, as in principle they are developed through ageostrophic flow in the proximity of the jet stream (Stohl et al., 2003). Deep folds extending down to the lower troposphere and occasionally to the ground surface may lead to irreversible mixing of stratospheric air into the troposphere and thus to chemical composition changes (Cristofanelli et al., 2006; Akritidis et al., 2010; Lin et al., 2015; Knowland et al., 2017). During recent years, several modelling studies indicated that the stratospheric contribution to tropospheric and near-surface background ozone may be of greater importance than previously anticipated (Zhang et al., 2011; Lin et al., 2012; Zanis et al., 2014; Lefohn et al., 2014; Akritidis et al., 2016; Williams et al., 2019).

Changes in ozone precursor emissions have the largest effect on future tropospheric ozone concentrations. Future reductions in most ozone precursor emissions, which are a common feature across the Representative Concentration Pathways (RCPs), drive tropospheric ozone decreases, except for RCP8.5, which shows an increase due to much larger methane concentrations compared to the other RCPs (Stevenson et al., 2006; Naik et al., 2013; Young et al., 2013; Sekiya and Sudo, 2014; Revell et al., 2015; Banerjee et al., 2016; Meul et al., 2018). Future decreases in ODS may lead to an ozone increase essentially everywhere in the atmosphere, with the largest percentage changes in the upper stratosphere and the lower stratosphere at high latitudes due to the anticipated ozone recovery, while changes in GHGs may lead to a decrease in the tropical lower stratosphere and an increase in STE due to strengthening of the BDC (Morgenstern et al., 2018). The 2014 Ozone Assessment CE2 (Carpenter et al., 2014) highlighted that chemistry–climate models (CCMs) robustly predict a long-term acceleration of the BDC in response to anthropogenic climate change (Hardiman et al., 2014; Palmeiro et al., 2014), which also applies for the new CCMI (Chemistry-Climate Model Initiative) simulations (Morgenstern et al., 2018).

Several recent studies with CCMs provide evidence that both the acceleration of the BDC and stratospheric ozone recovery will tend to increase the future global tropospheric ozone burden through enhanced STE with the magnitude of the change depending on the RCP scenario, partially off-

setting tropospheric ozone decreases associated with reductions in ozone precursor emissions (Sekiya and Sudo, 2014; Banerjee et al., 2016; Meul et al., 2018). Banerjee et al. (2016) showed that BDC strengthening under RCP8.5 has the largest impact on tropospheric ozone over the tropics and subtropics, while stratospheric ozone recovery from declining ODSs becomes more important in the mid-latitudes and extratropics. Meul et al. (2018) simulated that the global mean annual STT is projected to increase by 53 % between the years 2000 and 2100 under RCP8.5 and it will be smaller for RCP6.0, but the resulting relative change in the contribution of ozone with stratospheric origin to ozone in the troposphere is of comparable magnitude in both scenarios. The co-variability between STE and tropospheric ozone from observations was used to deduce that the projected future strengthening of the BDC alone (without accounting for ozone recovery), could lead to an increase in zonal-mean tropospheric ozone of 2 % by the end of the 21st century (Neu et al., 2014). Hess et al. (2015), extrapolating their model results from the present to future, concluded that a 30 % increase in the ozone flux by 2100 due to BDC strengthening would result in a 3 % increase in surface ozone and a 6 % increase in mid-tropospheric ozone. However, Morgenstern et al. (2018), using simulations from multiple CCMs, showed that the surface ozone response to anthropogenic forcings from well-mixed GHGs and ODSs remains uncertain, reflecting uncertainties related to STE.

There is a high confidence that the increasing temperature will lead to a decline of lower-tropospheric ozone through the enhanced water vapour abundances and the associated acceleration of ozone chemical loss (Fiore et al., 2012, 2015; Fu and Tian, 2019). Several studies indicate that the emissions of BVOCs are subject to increase in a warming climate, as they are temperature-sensitive, leading to a positive feedback on future ozone chemical production (Zeng et al., 2008; Weaver et al., 2009; Doherty et al., 2013). Yet, other studies considering the $CO_2$ inhibition effect report that this positive feedback on ozone may be offset or even reverse negatively (Tai et al., 2013; Hantson et al., 2017). Climate-related changes in lightning activity and the associated $NO_x$ emissions are thought to have complex implications for tropospheric ozone. While the enhancement in lightning $NO_x$ emissions in a warmer climate will increase baseline ozone, the induced enhancement in OH will result in $CH_4$ reduction and thus a decline of ozone chemical production on greater timescales (Wild, 2007; Banerjee et al., 2014; Murray, 2016). Moreover, climate-induced changes in $NO_x$ emissions from soils and ozone precursor emissions from wildfires are also expected to modulate future ozone changes (Voulgarakis and Field, 2015; Romer et al., 2018).

It is therefore crucial to conduct more studies on this topic in order to increase confidence in the future projected changes of tropospheric ozone and its associated drivers. This study aims to assess the impacts of future climate change under the RCP6.0 scenario on tropopause folds and

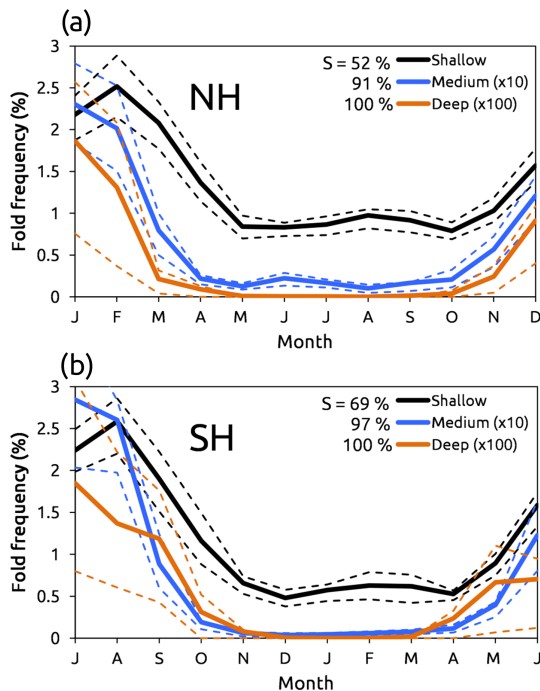

**Figure 1.** Seasonal cycle of tropopause fold frequencies (%) for **(a)** the NH (0–65° N) and **(b)** the SH (0–65° S) over the period 1979–2012 for intercomparison with Fig. 7 from Škerlak et al. (2015). The solid lines stand for the mean values, while the dashed coloured lines stand for the 25 % and 75 % percentiles. The seasonality $S = \frac{\max - \min}{\max + \min}$ of each seasonal cycle is also shown.

tropospheric ozone, using a free-running hindcast and projection ECHAM5/MESSy (EMAC) simulation for the period 1960–2100. To this end, a 3-D labelling algorithm is implemented to detect tropopause folds in EMAC simula-
5 tion. In addition to ozone, a tracer for stratospheric ozone is also employed to investigate the projected changes in STE of ozone. Section 2 presents the main characteristics of the EMAC model and describes the 3-D labelling algorithm used to detect the folding events. Sections 3 and 4 show the key re-
10 sults of the current study, and finally Sect. 5 summarizes the main conclusions.

## 2 Methodology

### 2.1 EMAC model

The ECHAM5/MESSy Atmospheric Chemistry (EMAC) global model is a numerical chemistry and climate simula-
15 tion system that includes sub-models describing tropospheric and middle atmosphere processes and their interactions with ocean, land, and human activities (Jöckel et al., 2010). It uses the second version of the Modular Earth Submodel System (MESSy2) to link multi-institutional computer codes.
The core atmospheric model is the Fifth generation circula-

tion model (ECHAM5; Roeckner et al., 2006). The EMAC model has been extensively evaluated for gas tracers (e.g. Pozzer et al., 2007; Jöckel et al., 2016) and for aerosols (e.g. Pringle et al., 2010; Pozzer et al., 2012; Astitha et al., 2012;
Pozzer et al., 2015). For the present study we use ECHAM5 version 5.3.02 and MESSy version 2.51. More specifically, data from the simulation RC2-base-04 are used, which is part of the set of simulations performed within the ESCiMo project (Jöckel et al., 2016) following the recommendations
by the CCMI. According to Eyring et al. (2013), the objective of REF-C2 (RC2) simulations is to produce best estimates of the future ozone and climate changes up to 2100, under specific assumptions about GHGs, as well as tropospheric ozone and aerosol precursors that follow RCP6.0,
and a specific ODS scenario that follows the halogen scenario A1 from WMO (2011b). The model horizontal resolution is T42L90MA, i.e. with a spherical truncation of T42 (corresponding to a quadratic Gaussian grid of approximately 2.8 by 2.8° in latitude and longitude) with 90 vertical hybrid
pressure levels up to 0.01 hPa.

The simulation covers the time frame 1960–2100 (10-year spin-up from 1950 to 1959) driven by prescribed sea surface temperature (SST) and sea ice coverage (SIC) taken from simulations with the global climate model HadGEM2-
45 ES (Collins et al., 2011; Martin et al., 2011) for the Coupled Model Intercomparison Project Phase 5 (CMIP5). Anthropogenic emissions are incorporated as prescribed emission fluxes following the CCMI recommendations (Eyring et al., 2013). In more detail, the emissions data set con-
50 sists of a combination of ACCMIP (Lamarque et al., 2010, for the 1950–2000 period) and RCP6.0 data (Fujino et al., 2006, for 2000 and on). Lightning $NO_x$ emissions and emissions of BVOCs are calculated online by the MESSy submodels $LNO_x$ (Tost et al., 2007) and ONEMIS (Kerkweg
et al., 2006), respectively, considering the effects of climate change. A detailed description of the simulation along with a comprehensive evaluation of ozone with satellite and ozonesonde measurements can be found in Jöckel et al. (2016, and references therein).
Along with ozone chemistry, EMAC also includes a tracer for ozone of stratospheric origin, denoted by O3s CE3, which provides an indicator of the stratospheric contribution to tropospheric ozone. In the stratosphere, O3s is equal to ozone values, while in the troposphere it follows the transport and
65 destruction processes of ozone. When O3s returns to the stratosphere it is reset to stratospheric values; however, since it is initialized above 100 hPa, only a very small fraction is recirculated by multiple crossings of the tropopause (Roelofs and Lelieveld, 1997).

### 2.2 Tropopause fold identification

In this work the algorithm developed by Sprenger et al. (2003) and improved by Škerlak et al. (2015) has been adopted and applied in order to detect tropopause folds in

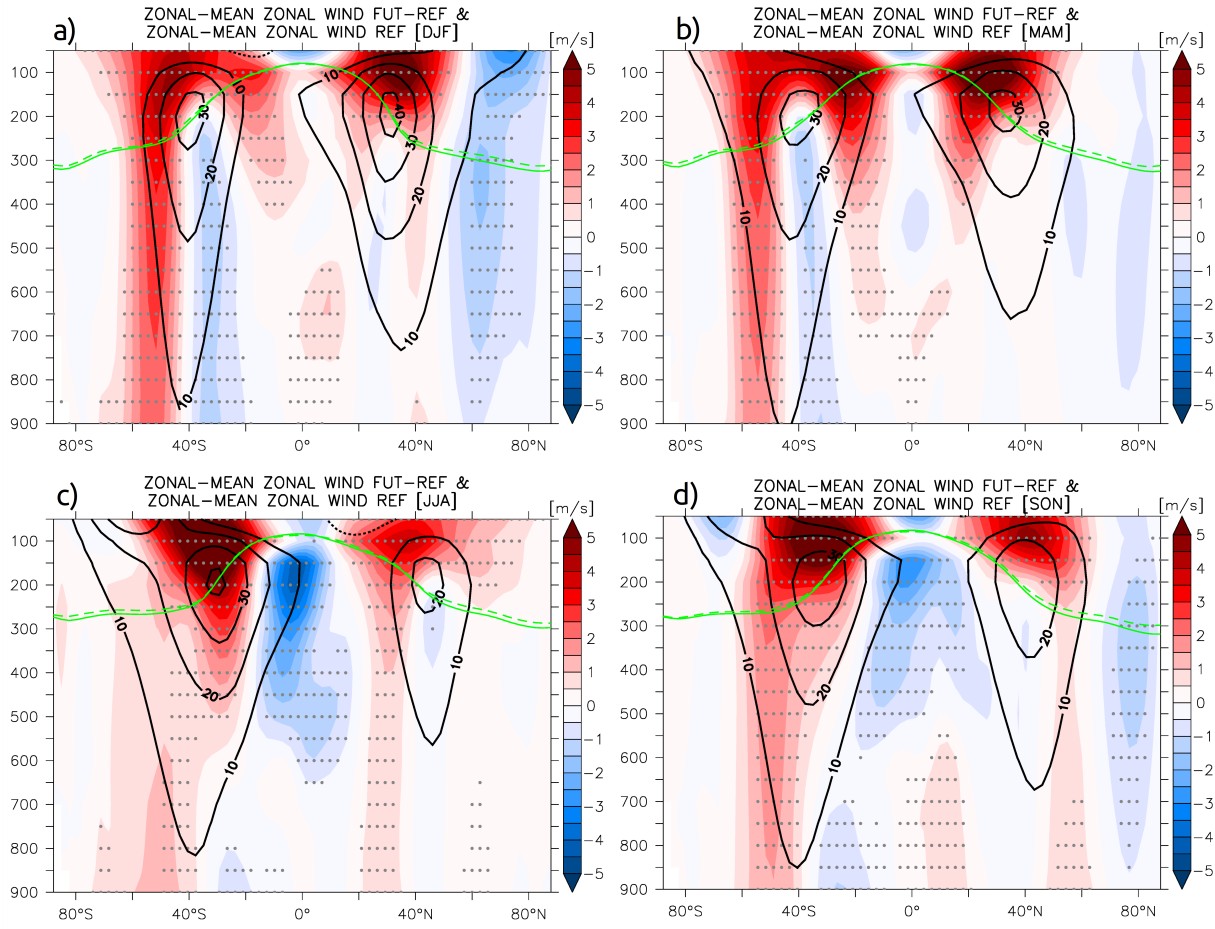

**Figure 2.** Zonal-mean zonal wind differences $(\mathrm{m\,s^{-1}})$ between the FUT and REF periods for DJF (**a**), MAM (**b**), JJA (**c**), and SON (**d**). The black contours indicate the zonal-mean zonal wind climatology $(\mathrm{m\,s^{-1}})$ for the REF period. The green solid (dashed) line denotes the height of the tropopause during the REF (FUT) period. Grey dots denote statistically significant changes at the 99 % confidence level.

EMAC simulation (as in Akritidis et al., 2016), using the 3-D fields of potential vorticity, potential temperature, and specific humidity. As in several previous studies (Hoskins et al., 1985; Holton et al., 1995; Stohl et al., 2003; Sprenger et al., 2003), the tropopause is defined as the combination of the isosurfaces of potential vorticity at $\pm 2$ PVU and potential temperature at 380 K, whichever is lower (referred to as dynamical tropopause). For each grid point a tropopause fold is designated where multiple crossings of the dynamical tropopause are detected in instantaneous vertical profiles. Subsequently, the upper ($p_\mathrm{U}$), middle ($p_\mathrm{M}$), and lower ($p_\mathrm{L}$) pressure levels of tropopause crossings are determined and the pressure difference $\Delta p = p_\mathrm{M} - p_\mathrm{U}$ between the upper and middle tropopause crossings is calculated (for more details see Fig. 1 in Tyrlis et al., 2014). The above pressure difference reveals the vertical extent of the tropopause fold and is used to classify the identified folds into three categories for more details see Škerlak et al. (2015):

- shallow folds, $50 \le \Delta p < 200$ hPa;

- medium folds, $200 \le \Delta p < 350$ hPa;

- deep folds, $\Delta p \ge 350$ hPa.

Before the results from simulation RC2-base-04 can be used to estimate the future projected changes of fold frequencies, the capability to reproduce present-time folding frequencies must be first checked. Therefore the model results have been compared with the monthly fold frequency climatology compiled by Škerlak et al. (2015). The climatology has been calculated using the same identification algorithm used in this work from the ERA-Interim dataset (Dee et al., 2011). Figure 1 shows the mean hemispheric (0–65° N and 0–65° S) monthly frequencies of different folding categories calculated from the results of simulation RC2-base-04 for the period 1979–2012, the exact same one covered by the work of Škerlak et al. (2015). This figure can be compared with Fig. 7 of Škerlak et al. (2015). The results are similar, implying a good representation of present-time monthly folding frequency. Yet, a small systematic overestimation of EMAC fold frequencies is seen. Additionally, not only the hemi-

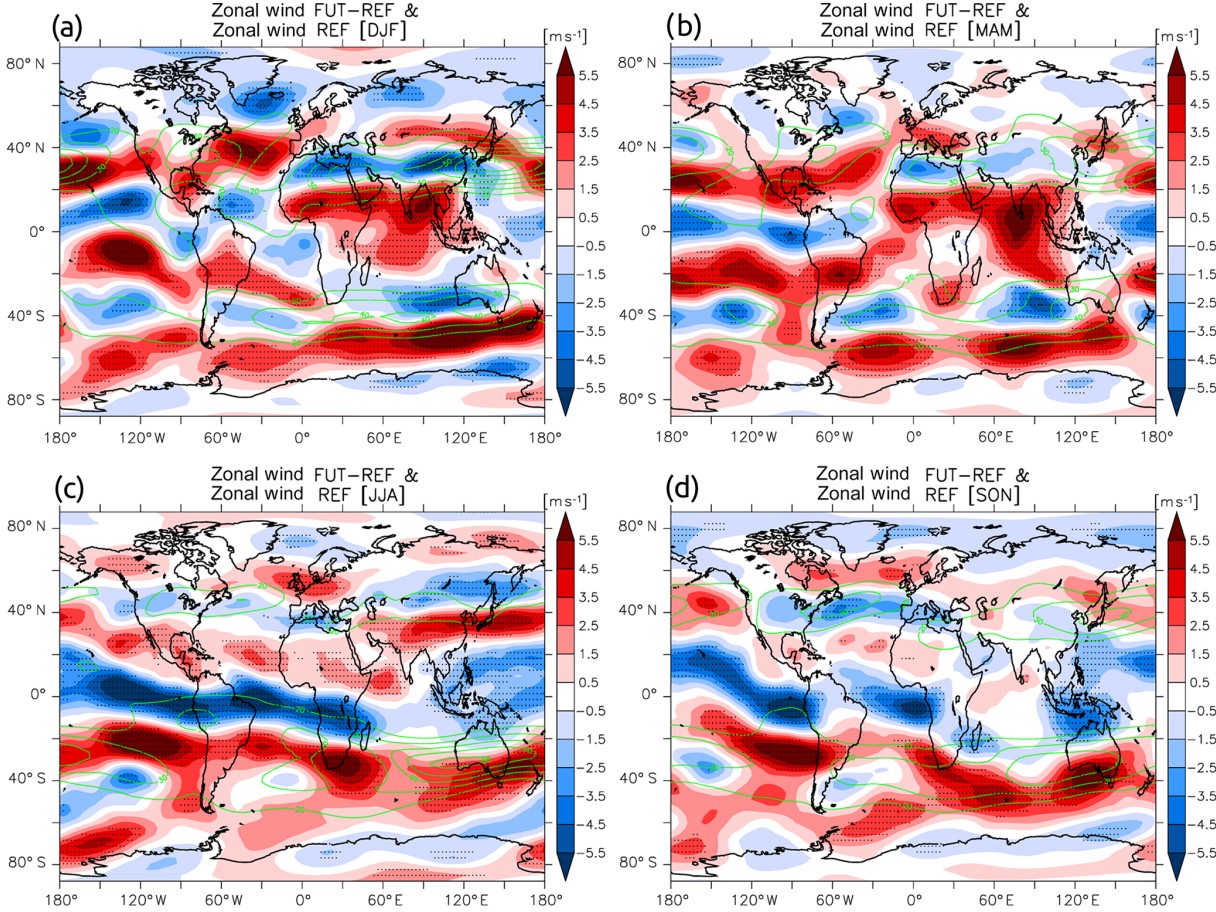

**Figure 3.** Mean zonal wind differences at 250 hPa (shaded; m s$^{-1}$) between the FUT and REF periods for DJF **(a)**, MAM **(b)**, JJA **(c)**, and SON **(d)**. The green contours represent the mean zonal wind at 250 hPa (m s$^{-1}$) during the REF period. The regions where the changes are statistically significant at the 99 % confidence level are hatched with black dots.

spheric monthly fold frequencies are similar between data from simulation RC2-base-04 and data from ERA-Interim; the geographical distribution also presents the same patterns (see Fig. 4). Any discrepancies might be attributed to the fact that RC2-base-04 is a free-running simulation with different horizontal and vertical resolution. We can therefore consider that the data used in this work are comparable for the present with state-of-the-art calculations based on the ERA-Interim dataset.

## 3 Future projected changes

To explore the future projected changes in EMAC meteorological and chemical parameters under the RCP6.0 emissions scenario, we consider two 30-year time periods: (a) present-day climate used as reference (REF) spanning from 1970 to 1999 and (b) future climate (FUT) spanning from 2070 to 2099. The selection of a 30-year period for the climate representation complies with the World Meteorological Organization's (WMO) suggestion (WMO, 2011a). All seasons in the

paper refer to boreal seasons (winter: DJF; spring: MAM; summer: JJA; autumn: SON).

### 3.1 Jet streams and tropopause folds

At first, the impact on atmospheric circulation under the RCP6.0 scenario is explored. As it is depicted from Fig. 2 there is a distinct upward and poleward shift in the Southern Hemisphere (SH) mid-latitude jet during all seasons, which is also identified during DJF and SON in the Northern Hemisphere (NH), yet less pronounced. A poleward–upward shift of the westerly jet in response to greenhouse warming was reported by several previous studies using individual models (Butler et al., 2010; Orbe et al., 2015; Doherty et al., 2017) or ensembles of models participating in the Intergovernmental Panel on Climate Change (IPCC) Fourth Assessment Report (Lorenz and DeWeaver, 2007), and the Coupled Model Intercomparison Project Phase 3 (CMIP3) and Phase 5 (CMIP5) (Swart and Fyfe, 2012; Delcambre et al., 2013; Yim et al., 2016). Moreover, a rise of the tropopause is seen during all seasons in both the NH and SH extratropics, which on an

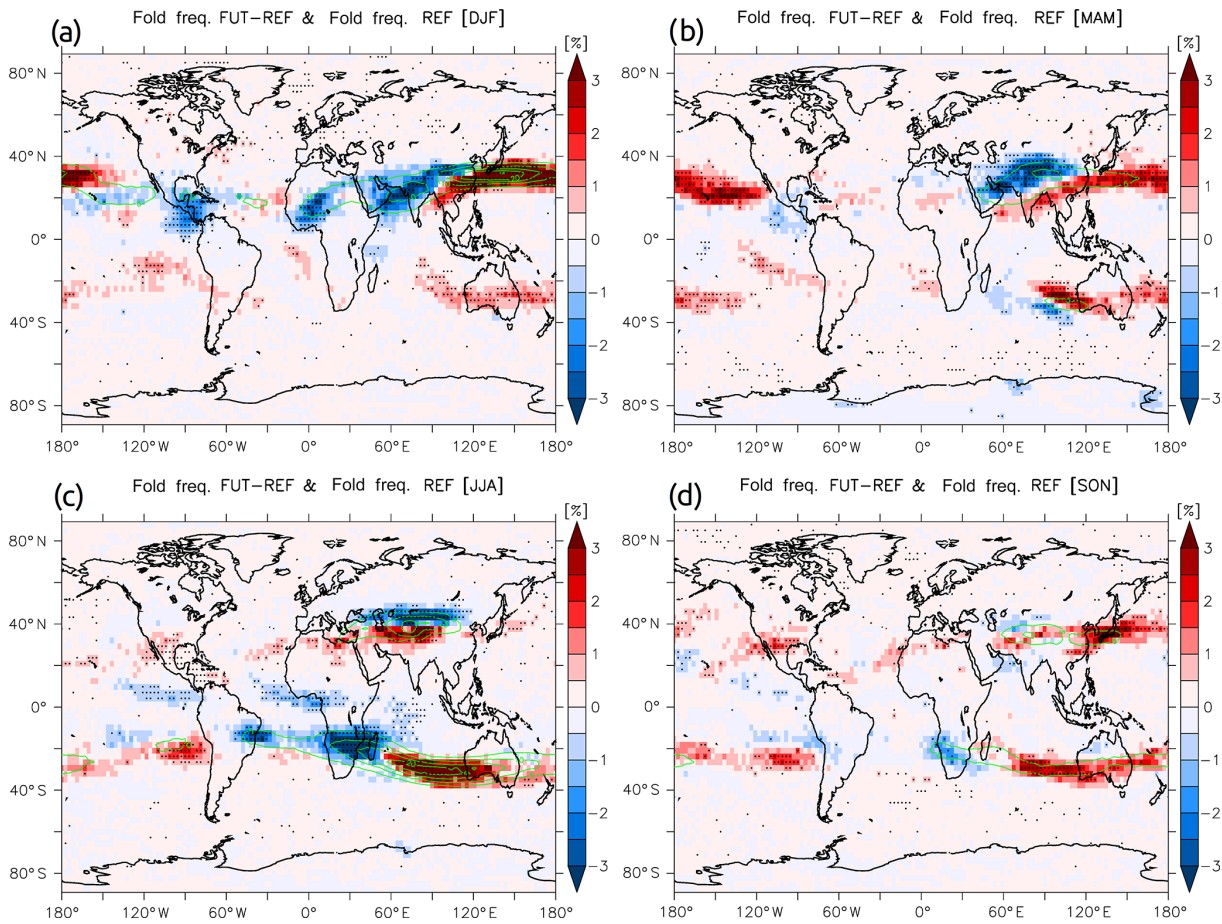

**Figure 4.** Mean tropopause fold frequency differences (shaded; %) between the FUT and REF periods for DJF **(a)**, MAM **(b)**, JJA **(c)**, and SON **(d)**. The green contours denote the tropopause fold frequency (%) during the REF period. The regions where the changes are statistically significant at the 99 % confidence level are hatched with black dots.

annual basis is estimated at about 8.8 and 5.8 hPa, respectively. A more comprehensive view of the present circulation patterns and their future changes is presented in Fig. 3. In the NH, a poleward shift of the zonal wind at 250 hPa is found during DJF over the Atlantic and eastern Asia, and an equatorward shift over the NH central and eastern Pacific is found, while in the SH a poleward shift is seen over the Indian Ocean. During JJA, an equatorward shift of the NH subtropical jet stream is depicted over central and eastern Asia, while in the SH a poleward shift is seen over Australia.

The impacts of the RCP6.0 emissions scenario on tropopause fold frequency are hereafter investigated, considering as folds all folds with $\Delta p \geq 50$ hPa (shallow, medium, and deep). Figure 4 presents the projected fold frequency changes between the FUT and REF periods along with the climatology of fold frequencies during the REF period for every season. The spatial distribution of fold frequencies during the REF period (green contours in Fig. 4) indicates that in principal folds occur in the regions with high zonal wind speed (green contours in Fig. 3). Noteworthy are the hotspots

during the REF period over Asia and the Middle East during DJF and JJA and over the southern Indian Ocean during JJA, whereas during the transition seasons the maxima are located over Asia in MAM and over Asia and the southern Indian Ocean in SON, which is consistent with the ERA-Interim-derived tropopause fold climatology of Škerlak et al. (2015). The projected changes in fold occurrence for the FUT period with respect to the REF period during DJF reveal a distinct pattern of decrease (increase) in fold frequency over south Asia (NH Pacific Ocean), associated with the adjacent decrease (increase) in zonal wind in the upper troposphere depicted in Fig. 3a. During JJA, the equatorward shift of the subtropical jet stream over central Asia implies a dipole pattern of decrease–increase in fold frequencies, while in the SH a decrease (increase) in fold occurrence is found over southern Africa (Indian Ocean) as a response to the projected changes in the upper-tropospheric zonal winds. During MAM a distinct increase in fold frequency prevails in a zone extending across the NH Pacific Ocean, and a decrease prevails in the north of India, while during SON more fre-

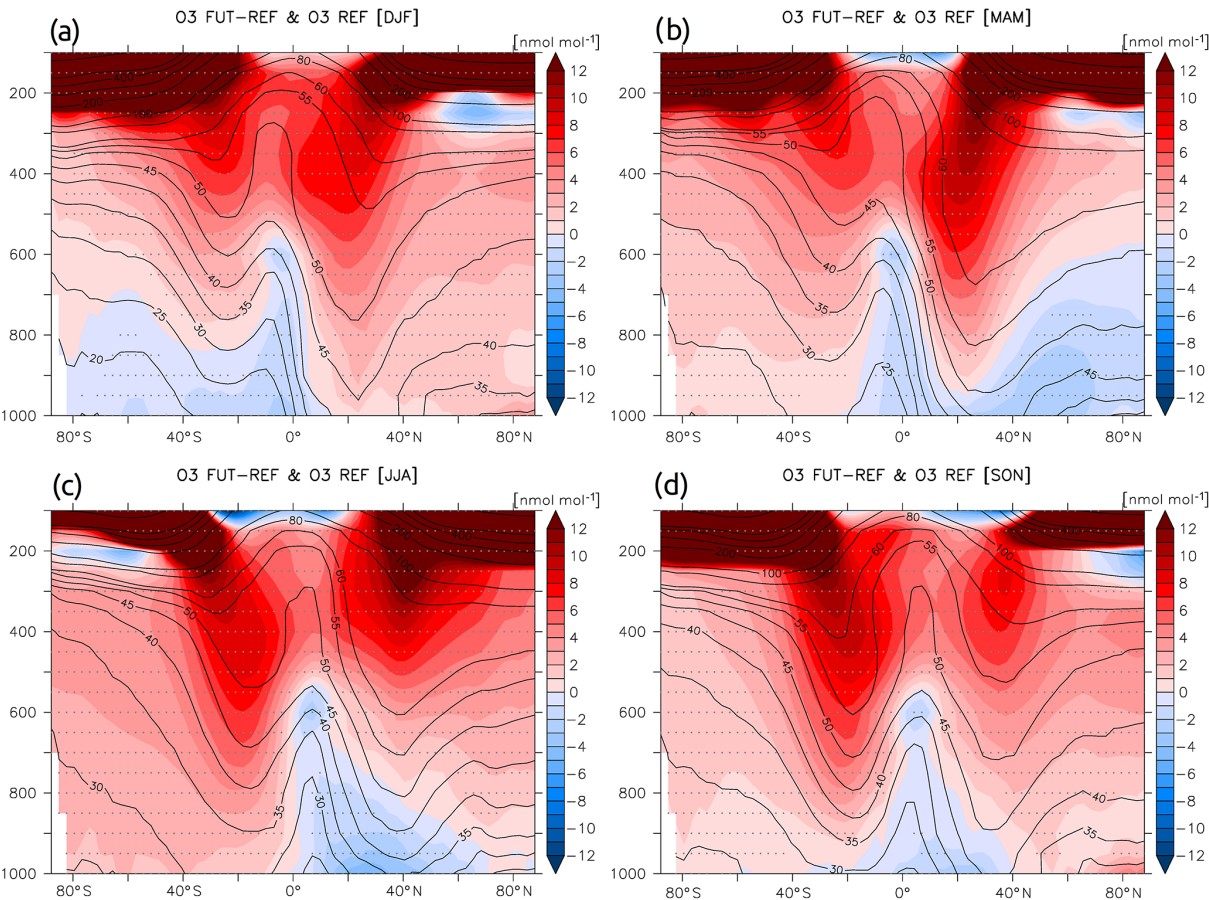

**Figure 5.** Differences of zonal-mean ozone concentrations between the FUT and REF periods (shaded; nmol mol$^{-1}$) for DJF **(a)**, MAM **(b)**, JJA **(c)**, and SON **(d)**. The black contours denote the zonal-mean ozone concentrations (nmol mol$^{-1}$) during the REF period. Grey dots denote statistically significant changes at the 99 % confidence level.

quent folding events are projected over the NH western Pacific Ocean and the Indian Ocean.

## 3.2 Tropospheric ozone

Here we explore the future changes in tropospheric ozone under the RCP6.0 GHG scenario. Figure 5 presents the projected changes of zonal-mean ozone along with its climatological values during the REF period on a seasonal basis. The highest concentrations of zonal-mean ozone in the troposphere during the REF period are found in the NH mid-latitudes during MAM and JJA and in the SH mid-latitudes during SON. With respect to the REF period, a decrease in zonal-mean ozone in the lower troposphere of up to 3 nmol mol$^{-1}$ is projected for the FUT period during MAM and JJA in the NH, and similarly during DJF (austral summer) in the SH, resulting from the RCP6.0 future ozone precursor emissions reduction, which as expected dominates during the seasons with more intense photochemistry. Additionally, even if we have no change in precursor emissions, as has been also outlined in the IPCC Fifth Assessment Report

(Kirtman et al., 2013), there is high confidence that in unpolluted regions, higher water vapour abundances and temperatures in a warmer climate enhance ozone destruction, leading to lower baseline ozone levels, while there is medium confidence that in polluted regions it is expected to increase surface ozone. This is also the case in the examined simulation, as the projected increase in water vapour mixing ratios contributes to the decrease in lower-tropospheric ozone through its enhanced chemical loss (not shown). Clearly, temperature and humidity under a warmer climate play an important role in decreasing tropospheric ozone in the tropical Pacific, due to the increased rate of the ozone destruction reactions (Revell et al., 2015). The aforementioned decreases in lower-tropospheric ozone are overcoming the appearing increases in ozone chemical production (not shown), which are likely associated with the enhanced emissions of BVOCs and lightning NO$_x$ (see Figs. 2, 3, and 4 in Jöckel et al., 2016). On the contrary, in the extratropical lower stratosphere and the upper and middle troposphere ozone is projected to increase during all seasons. The largest increases in the upper and middle troposphere, of up to 12 nmol mol$^{-1}$, are seen in the subtropics

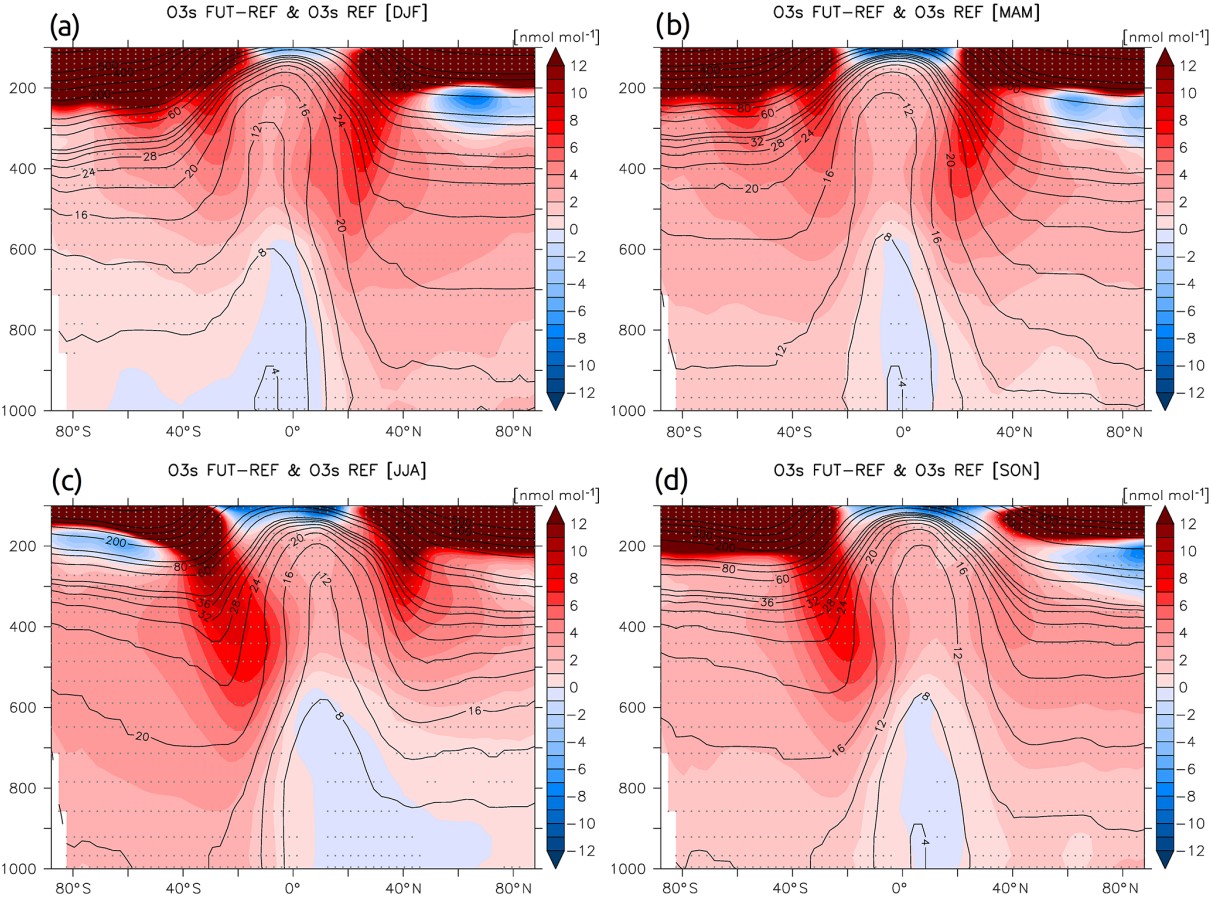

**Figure 6.** Differences of zonal-mean stratospheric ozone tracer (O3s) concentrations between the FUT and REF periods (shaded; nmol mol$^{-1}$) for DJF (**a**), MAM (**b**), JJA (**c**), and SON (**d**). The black contours denote the zonal-mean stratospheric ozone tracer concentrations (nmol mol$^{-1}$) during the REF period. Grey dots denote statistically significant changes at the 99 % confidence level.

and in the vicinity of the jet streams where tropopause fold formation and the induced STT are favoured. The more pronounced increases of ozone are found in the NH (SH) during MAM (SON) throughout the entire free troposphere. These patterns of tropospheric ozone increase are due largely to a global STT increase, linked to stratospheric ozone recovery and a strengthening of BDC, as suggested by previous studies based on simulations with CCMs (Banerjee et al., 2016; Morgenstern et al., 2018). The enhanced lightning NO$_x$ values are also likely to be auxiliary in the direction of increasing tropospheric ozone. In the free troposphere, it seems that the beneficial reduction of ozone precursor emissions and the ozone decline due to higher water vapour content are cancelled out by the projected increase in stratospheric ozone influx and ozone chemical production from BVOC and lighting NO$_x$. In regards to lower stratosphere, an increase in ozone is projected outside the tropics reflecting the recovery of stratospheric ozone. In the tropical lower stratosphere, the projected decrease in ozone is presumably related to the BDC strengthening and the induced increased upwelling of tropospheric ozone-poor air into the lower stratosphere. This trop-

ical lower-stratospheric ozone decrease under an increase in GHGs, due to a BDC strengthening and the induced upwelling enhancement, has been reported in other studies as well (e.g. Zeng et al., 2010; Young et al., 2013; Banerjee et al., 2016; Abalos et al., 2017). Specifically, Abalos et al. (2017), using the artificial tracer e90, suggested an increase in the tropical upwelling and thus a stronger vertical TST in the future.

### 3.3 Stratospheric ozone tracer (O3s)

To estimate the impact of STE on tropospheric ozone, the projected changes of O3s are examined here. Same as in Fig. 5, Fig. 6 depicts the differences of zonal-mean O3s concentrations between the FUT and REF periods. An increase in O3s occurs almost throughout the troposphere during all seasons. In the NH, the peak of O3s enhancement is found in the subtropics and in the vicinity of the NH jet stream during DJF and MAM (Fig. 6a and b), while in the SH the respective positive maxima are seen during JJA and SON (Fig. 6c and d), similarly near the position of the SH jet stream.

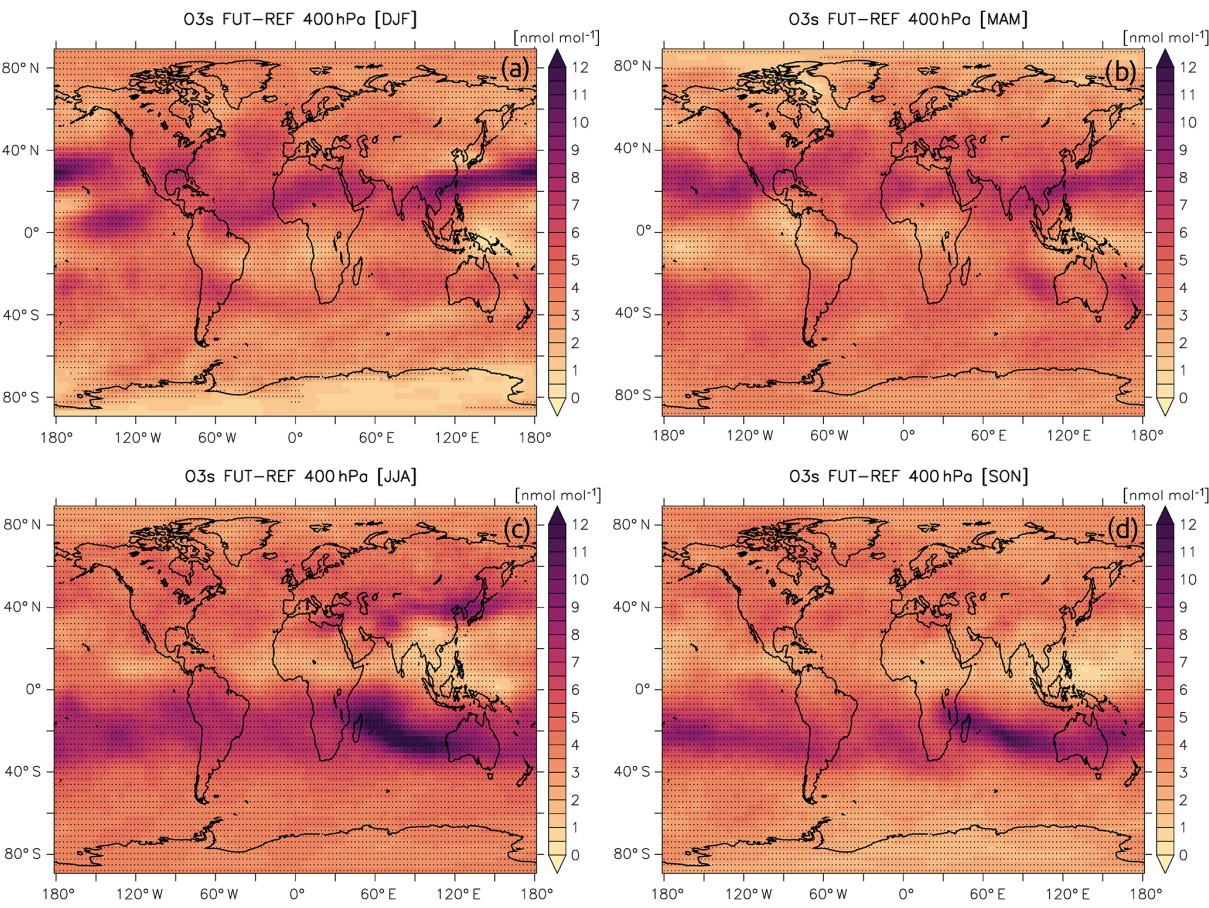

**Figure 7.** Mean stratospheric ozone tracer (O3s) concentration differences (shaded; nmol mol$^{-1}$) between the FUT and REF periods at 400 hPa for DJF **(a)**, MAM **(b)**, JJA **(c)**, and SON **(d)**. Black dots denote statistically significant differences at the 99 % significance level.

These increases in O3s in the NH–SH subtropical upper and middle troposphere reveal an increase in isentropic cross-tropopause ozone transport, through tropopause folds that in principal occur near the NH–SH subtropical jet streams. In general, the positive O3s patterns resemble those of tropospheric ozone (Fig. 5), indicating that the projected increase in tropospheric ozone is largely driven by the increase in STT and the induced vertical transport of stratospheric ozone in the underlying troposphere, as was also reported from previous modelling studies employing a tracer for stratospheric ozone in future projected sensitivity simulations (Banerjee et al., 2016; Meul et al., 2018). Meul et al. (2018) in their future projected simulations under the RCP8.5 GHG scenario with the EMAC model noted a similar increase in ozone STT through the strengthening of the BDC and the increase in the net ozone production in the stratosphere, which was attributed to the rising GHG concentrations. A small decrease in O3s occurring mainly in the SH (NH) lower troposphere during DJF (JJA) is associated with an increased chemical O3s loss due to a slight increase in OH and HO$_2$ and their reaction rate with ozone (due to increased temperature).

The spatial distribution of O3s projected changes at 400 hPa is presented in Fig. 7, to identify the global hot spots of climate change impact on ozone STT. Overall, an increase in ozone with stratospheric origin is projected in the middle troposphere (400 hPa) during all seasons, reflecting the recovery of stratospheric ozone and the associated increase in ozone STE. Notably, the maxima of O3s increase coincides mainly with the respective maxima of tropopause fold frequency increase (see Fig. 4). In more detail, during DJF the peaks of future O3s increases (up to 12 nmol mol$^{-1}$) are found over the NH Pacific Ocean (Fig. 7a), while during JJA the respective peaks (exceeding 12 nmol mol$^{-1}$) mainly occurred over central Asia and the Indian Ocean (Fig. 7c). All in all, the emerging increase in ozone STE under the RCP6.0 GHG scenario is mostly driven by the strengthening of BDC and the recovery of stratospheric ozone; still for regions where tropopause folds are projected to occur more often, the downward transport of ozone from the stratosphere seems to be more pronounced.

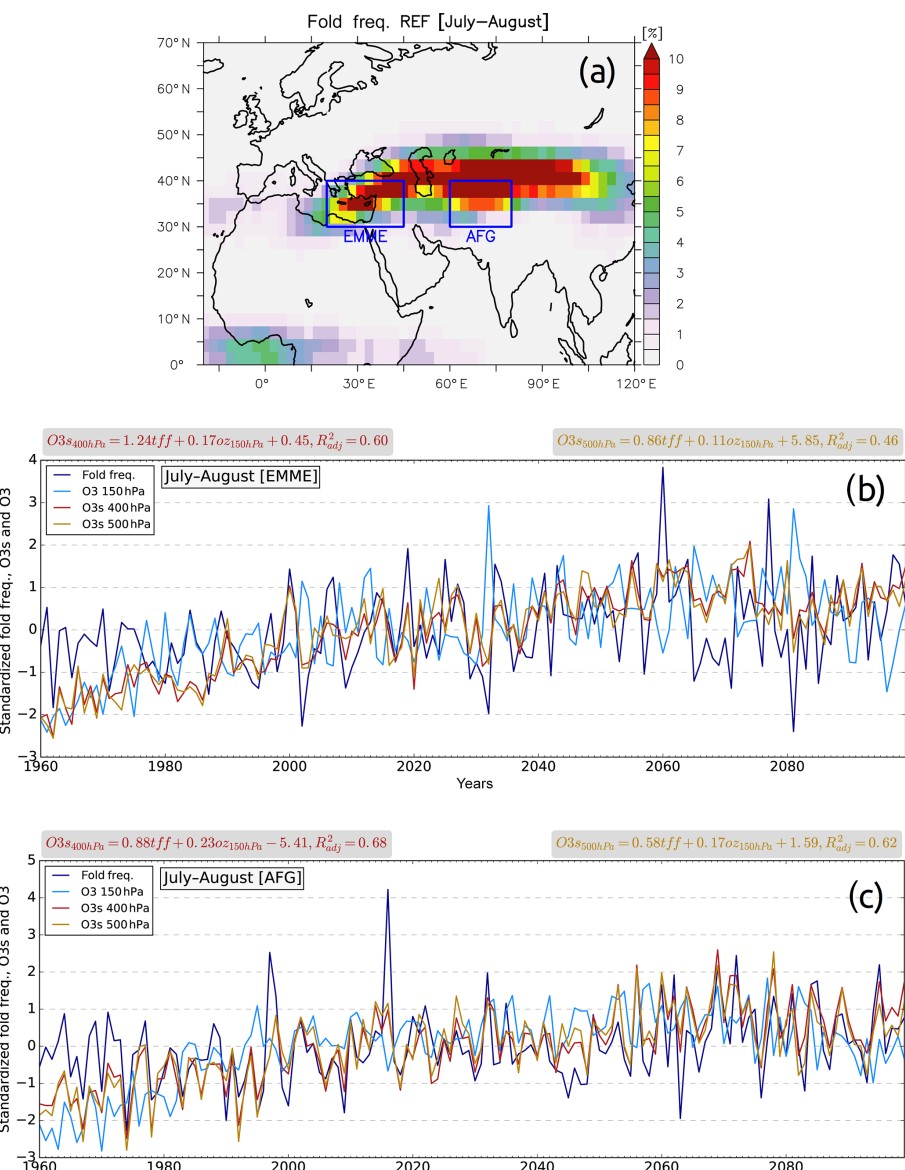

**Figure 8.** Mean July–August tropopause fold frequency (%) during the REF period and the examined EMME (20–45° E, 30–40° N) and AFG (60–80° E, 30–40° N) regions **(a)**. Time series of standardized mean July–August tropopause fold frequency (dark blue line), O3 at 150 hPa (light blue line), O3s at 400 hPa (dark red line), and O3s at 500 hPa (orange line) over EMME **(b)** and AFG **(c)** for the period 1960–2100. Regression equations for O3s at 400 and 500 hPa are also shown at the top of the charts with dark red and orange, respectively.

## 4 Hot spots of ozone STT

STT is of great importance for ozone levels and variability in the upper–middle troposphere over regions where the meteorological conditions favour the formation of tropopause folds and downward transport (Roelofs and Lelieveld, 1997; Sprenger and Wernli, 2003), such as the eastern Mediterranean and the Middle East (EMME) (Li et al., 2001; Zanis et al., 2014; Akritidis et al., 2016) and the broader Afghanistan area (AFG) (Tyrlis et al., 2014; Ojha et al., 2017) during summer and especially the July–August pe-

riod. To explore the links of the tropopause fold frequency and stratospheric ozone with the interannual variability of middle-tropospheric ozone with stratospheric origin over the EMME (20–45° E, 30–40° N) and AFG (60–80° E, 30–40° N) regions, the mean July–August time series of tropopause fold frequency, ozone at 150 hPa, and O3s at 400 and 500 hPa for the period 1960–2099 were constructed. Figure 8a presents the mean July–August fields of tropopause fold frequency during the REF period, revealing a pronounced fold activity over the depicted EMME and AFG regions. For the EMME region (Fig. 8b), the interannual

variability of mean July–August O3s at 400 hPa (500 hPa) is found to be positively correlated at the 99 % significance level with the mean July–August tropopause fold frequency and ozone at 150 hPa, with values of $r = 0.53$ ($r = 0.43$) and $r = 0.56$ ($r = 0.49$), respectively. Employing a multiple linear regression analysis, fold frequency and ozone at 150 hPa are found to explain the 58 % (42 %) of the variance of O3s at 400 hPa (500 hPa). With regards to the AFG region, the variance of the projected mean July–August O3s concentrations at 400 hPa (500 hPa) explained by fold frequency and ozone at 150 hPa is 73 % (68 %). The year-to-year variability of July–August O3s at 400 hPa (500 hPa) is found to be positively correlated at the 99 % significance level with both fold frequency $r = 0.64$ ($r = 0.58$) and ozone at 150 hPa $r = 0.64$ ($r = 0.64$).

## 5 Conclusions

This study investigates the future projected changes in tropopause folds, ozone STT, and tropospheric ozone under the RCP6.0 emissions scenario, using a transient simulation with the EMAC chemistry–climate model (CCM) CE4 from 1960 to 2100 and a tropopause fold identification algorithm. In particular, we examined the long-term change in tropopause fold frequency and the potential links with atmospheric circulation changes. Moreover, the long-term changes in tropospheric ozone and ozone STT were also explored and associated with the respective variations in fold activity. The most noteworthy findings of the present study can be summarized as follows.

- Robust changes in atmospheric circulation are identified under the RCP6.0 GHG emissions scenario. A poleward and upward shift of the NH subtropical jet is projected for DJF and SON, while a strengthening of zonal-mean wind in the upper troposphere is seen equatorward for JJA. The responses are more pronounced in the SH, showing a distinct poleward shift for DJF and MAM, with a strengthening of zonal-mean wind poleward during JJA and SON.

- The spatial patterns of the projected changes in NH and SH subtropical jets seem to drive the respective patterns of future tropopause fold frequency changes, with a negative–positive dipole structure found over south Asia and the NH Pacific Ocean during DJF and MAM. The most prominent features during JJA are a distinct increase in fold activity over the Indian Ocean exceeding 3 % and a negative–positive dipole structure centred over the greater Afghanistan region.

- The regions exhibiting the highest increases in tropopause fold occurrence in the future are those with the more pronounced projected increases in O3s in the middle troposphere (400 hPa). The projected changes

of zonal-mean O3s concentrations reveal a strengthening of ozone STT at the middle latitudes of both hemispheres during all seasons, which is more distinct in the NH during DJF and MAM (up to 6 nmol mol$^{-1}$ down to 500 hPa) and in the SH during JJA and SON (up to 8 nmol mol$^{-1}$ down to 500 hPa). Although the future increase in ozone STT on a global scale seems to be forced from stratospheric ozone recovery and strengthening of BDC (Banerjee et al., 2016; Meul et al., 2018), regionally, the degree of increase in the downward transport of stratospheric ozone is partially driven by the long-term changes in fold activity.

- For specific regions considered to be global STT hotspots, namely the summertime EMME and AFG, the projected year-to-year variability of middle-tropospheric ozone with stratospheric origin seems to be largely governed by both the variabilities of ozone at 150 hPa and fold frequency, as they explain 60 % and 68 % of the variance of mean July–August O3s concentrations at 400 hPa for EMME and AFG, respectively, over the period 1960–2100.

- Ozone in the lower troposphere and near the surface decreases under the projected decline in ozone precursor emissions and the effect of increased water vapour content. In the middle and upper troposphere the projected strengthening of ozone STT contributes to the increase in ozone globally.

In summary, the findings of this study are in the same direction as other studies based on different CCMs (Zeng et al., 2010; Banerjee et al., 2016; Meul et al., 2018), increasing confidence in the direction of an increased ozone STT and induced increases in middle- and upper-tropospheric ozone in the future under the RCP6.0 emissions scenario. The role of tropopause fold activity in a changing climate seems to be a considerable factor for both the levels and variability of ozone STT.

*Data availability.* .TS4

*Author contributions.* DA performed the analysis and wrote the paper with contributions from PZ and AP. AP provided the EMAC model data. PZ and AP contributed to the interpretation of the results.

*Competing interests.* The authors declare that they have no conflict of interest.

*Special issue statement.* This article is part of the special issue "The Modular Earth Submodel System (MESSy) (ACP/GMD interjournal SI)". It is not associated with a conference. TS5

*Acknowledgements.* The EMAC model simulations were performed at the German Climate Computing Center (DKRZ) with support from the Bundesministerium für Bildung und Forschung (BMBF). The authors gratefully acknowledge DKRZ and its scientific steering committee for providing the HPC and data archiving resources for the consortial project ESCiMo (Earth System Chemistry integrated Modelling). The authors are furthermore grateful to Patrick Jöckel for his contribution to the ESCiMo simulations and the EMAC model development. The authors also acknowledge Michael Sprenger (ETH Zurich) for the development of the 3-D labelling algorithm, which is used in the present study for the detection of tropopause folds in EMAC simulation. Dimitris Akritidis acknowledges the Research Committee of the Aristotle University of Thessaloniki (https://www.rc.auth.gr TS6) for the 2015 Postdoctoral Excellence Fellowship.

*Review statement.* This paper was edited by Pedro Jimenez-Guerrero and reviewed by three anonymous referees.

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

TS23    Please add page range or article number.

TS24    Please add page range or article number.

TS25    Please add total pages.

TS26    Please add page range or article number.