# Peer review of "On the impact of future climate change on tropopause folds and tropospheric ozone"

_Atmospheric Chemistry and Physics, 2019_

## Referee Comment (RC1) · Anonymous Referee #1 · 10 Jul 2019

General comment:

This manuscript by Akritidis et al. analyzes the impact of future changes in the tropopause fold frequency on concentrations of tropospheric ozone. The authors use an atmospheric chemistry global model and a well-known tropopause fold identification algorithm, to analyze variations in the stratosphere-to-troposphere transport (STT) of ozone, under the RCP6.0 scenario.

The study is certainly of interest, since the topic of stratosphere-to-troposphere exchange (STE) is of great importance, especially for what concerns the future ozone variations, which would naturally undergo a decrease in the lower troposphere, as projected by precursors emissions reduction.

This is an interesting study and a well written paper, and I recommend publication

in ACP after addressing the comments listed below. In particular, the study could be more complete if also the role of troposphere-to-stratosphere transport (TST) is taken into account, especially to quantify whether the ozone reduction in the middle and upper troposphere (due to precursors emissions reduction) is "overcome" by the increase in ozone due to STT, which seems to occur globally.

Specific comments:

1. Page 3, Line 7. The authors should motivate the choice of the RCP6.0 scenario. Apart from the RCP8.5, which was already assessed in the past, why not choosing, e.g., RCP2.6 or RCP4.5?

2. Page 4, Lines 23–28. Do the authors take into account any limitations of the work by Škerlak et al. (2015)? How would these affect the comparison between the two methodologies?

3. Page 6, Lines 14–23. The strengthening of the BDC would imply more rising air in the tropics, which would then be reflected in a decrease of ozone in the tropical lower stratosphere. Is there any evidence on this, also based on TST (troposphere-to-stratosphere transport) studies? In particular, is Line 19 ("increased upwelling of tropospheric ozone-poor air into the lower stratosphere"), supported by any result? At line 20, the authors indicate a "global STE increase" as the main cause of tropospheric ozone increase, but would this include an increase in both of the two components, i.e., STT and TST, or does it refer to STT only?

4. Page 7, Lines 2–3. In which way is the increase in GHGs concentrations related to the increase in STE of ozone?

5. Page 7, Lines 12–14. Again, the role and quantification of TST in not taken into

account here. What role would it play in modulating the increase of ozone STE reported in the paper?

6. Page 8, Lines 28–31. Would it be possible to "quantify" the effect of these two contributions (i.e., reduction of ozone precursors emissions and increase of ozone STT), so that one could quantitatively see that the ozone decrease due to emissions reduction is effectively canceled out by the global ozone increase due to STT?

Technical corrections:

1. Page 5, Line 21. "Green contours", please revise Fig. 4 caption, i.e., "black"→"green".

2. Pag. 6, Lines 28–29. Please check correspondence between Figure numbering and seasons.

3. Figure 7. "concnentrations"→"concentrations".

4. Page 7, Line 21. "EM" or "EMME"? Please be consistent.

5. Page 7, Lines 25 and 30. "positevely"→"positively".

---

## Referee Comment (RC2) · Anonymous Referee #2 · 12 Jul 2019

The authors explore the roles of future climate change in tropospheric ozone changes using a global chemistry-climate model with artificial stratospheric ozone tracer. The results of this study emphasize the importance of downward transport of stratospheric ozone associated with tropopause folds. You've convinced me that changes in tropopause folds are regulated by upper-level jet. Also, I agree that projected increase of tropospheric ozone is associated with changes in BDC and STT. However, I find the linkage between the presence of folds and changes in ozone is relatively weak. I would expect shallow tropopause folds, which are located above 200hPa, account for the most changes in folding frequency. How do these shallow folding activities affect the ozone near 400-500hPa or even below? We know that summertime large-scale subsidence at 500hPa over Mediterranean is projected to change [Cherchi et al., Clim

Dyn (2016)]. Perhaps the large changes in ozone near 400hPa are primarily associated with changes in descent, while the presence of tropopause folds is secondary. Except the one concern I've pointed out, this paper is very well structured and is certainly within the scope of ACP, although improvements can be applied to make it clearer. Therefore, I only have some minor comments.

1. What's your rationale for choosing RCP6?

2. Ozone is difficult to simulate in models due to biases in photochemistry processes and precursor emissions. Have you evaluated model performance in ozone? Discussion regarding how biases in EMAC would affect the estimated changes is necessary.

3. In Fig.1, wintertime medium and deep fold frequency are much higher than those shown in Škerlak et al. (2015). Will it affect your results? Also, it'd be good to address that the climatological distribution of tropopause folds in your model is consistent with what shown in previous studies.

4. I'm worried whether the future changes of tropopause folds are robust. Have you compared with other models?

Detailed comments:

1. P18, Fig.4 caption, "black" -> "green"; "circles" -> "dots"

2. P6, line 13, "contrary" -> "on the contrary"

3. P7, "positevely" -> "positively"

---

## Referee Comment (RC3) · Anonymous Referee #3 · 22 Jul 2019

General comments

This is an interesting and well-presented paper. My main concern is that like many of the past papers that discuss the influence of climate change (either past or future) on tropospheric ozone, it is difficult to separate out the individual effects of different processes. You appear to be assuming that all (or the vast majority) of your signal is just the combination of changes in STT, together with changes of anthropogenic emissions under RCP6.0. But what about changes in water vapour, and natural emissions from lightning NOx and BVOCs (etc.)? Most of these are barely discussed in the paper, but I think they must be simultaneously changing, and having potentially large effects. Some authors have attempted to separate out some of these processes in the past (e.g., Wild, 2007; Doherty et al.,2013), but this is not easy. This wider context needs to

be discussed to place some perspective on where changes in the STT rank compared to other climate change effects on ozone. If this can be included, and the points below, then I am happy to recommend publication in ACP.

Specific comments

P1

L4: Clarify the temporal and spatial context of the 3% increase (i.e. from (1970-99) to (2070-99); is it a global average number, or related to the model grid size?)

L8: maxima -> largest

L9: Highest background fold frequencies, or changes?

Abstract: How is the (likely) shortened lifetime of tropospheric ozone in future, due to higher levels of water vapour, and hence bigger flux through $O(1D)+H_2O$, taken into account? Also what about changes in lightning $NO_x$ emissions (and BVOC emissions, and other climate dependent processes...) that may affect tropospheric $O_3$? Introduction: This should also mention other climate-driven influences on tropospheric $O_3$ – ie water vapour, lightning $NO_x$, biogenic VOC emissions, etc.

P4

L1: Presumably the stratospheric ozone tracer ignores rapid cycling processes involving $O_3$, ie.: $O_3 + NO \rightarrow NO_2 + O_2$ $NO_2 + h\nu \rightarrow NO + O$ $O_2 + O \rightarrow O_3$ Which form a null cycle. But presumably it does include $O_x$ ($O_3+NO_2$) loss processes that interact with this cycle, such as $O(1D)+H_2O$ and $NO_2$ dry deposition?

P5

L7 in the Northern Hemisphere, not at the Northern Hemisphere (and several other similar instances). 'At' is appropriate for a specific site, whereas 'in' is more appropriate for a larger region. I don't think this is just my dubious grammar.

L22 Do you mean the hotspots in the REF distribution, or the changes?

L26 delete 'a'

L27 It is a bit confusing that Figure 3 has colours for REF winds and contours for FUT-REF changes, whilst Figure 4 has contours for REF fold frequencies and colours for changes. I suggest all the figures follow a consistent format?

P6

L12 lower tropospheric ozone?

Section 3.2 What about lightning NOx? Does it change? And BVOCs?

Section 3.3 From your experiments it is not possible to separate the effects of stratospheric O3 recovery (due to ODS declines) and enhanced STE. Is that correct?

P7

L25 and l30 positively

Section 4: Should EM be EMME?

P18, Figure 4 caption: green not black. What are the units of fold frequency? "hatched with black circles" -> "indicated by black dots".

References

Doherty, R. M., et al. (2013), Impacts of climate change on surface ozone and intercontinental ozone pollution: A multi-model study, J. Geophys. Res. Atmos., 118, doi:10.1002/jgrd.50266

Wild, O.: Modelling the global tropospheric ozone budget: exploring the variability in current models, Atmos. Chem. Phys., 7, 2643-2660, https://doi.org/10.5194/acp-7-2643-2007, 2007.

[Figure]

2019.

---

## Author Comment (AC1) · 3 Oct 2019

Note: Reviewer's comments are presented in black font; authors' responses are presented in blue plain font; manuscript text quotations are presented in blue italics font.

Anonymous Referee #1

We would like to thank Reviewer #1 for her/his time devoted and the constructive and helpful comments.

General comment:

This manuscript by Akritidis et al. analyzes the impact of future changes in the tropopause fold frequency on concentrations of tropospheric ozone. The authors use an atmospheric chemistry global model and a well-known tropopause fold identification algorithm, to analyze variations in the stratosphere-to-troposphere transport (STT) of ozone, under the RCP6.0 scenario. The study is certainly of interest, since the topic of stratosphere-to-troposphere exchange (STE) is of great importance, especially for what concerns the future ozone variations, which would naturally undergo a decrease in the lower troposphere, as projected by precursors emissions reduction. This is an interesting study and a well written paper, and I recommend publication in ACP after addressing the comments listed below. In particular, the study could be more complete if also the role of troposphere-to-stratosphere transport (TST) is taken into account, especially to quantify whether the ozone reduction in the middle and upper troposphere (due to precursors emissions reduction) is "overcome" by the increase in ozone due to STT, which seems to occur globally.

We thank the Reviewer for the comments, to which we will respond point by point.

Specific comments:

1. Page 3, Line 7. The authors should motivate the choice of the RCP6.0 scenario. Apart from the RCP8.5, which was already assessed in the past, why not choosing, e.g., RCP2.6 or RCP4.5?

The examined simulation RC2-base-04 is part of the Earth System Chemistry integrated Modelling (ESCiMo) initiative chemistry–climate simulations, which have been conducted by the MESSy Consortium with the EMAC model following the recommendations by the Chemistry-Climate Model Initiative (CCMI). According to Eyring et al. (2013), the objective of REF-C2 (RC2) is to produce best estimates of the future ozone and climate changes up to 2100, under specific assumptions about GHG as well as tropospheric ozone and aerosol precursors that follow RCP 6.0 and a specific ODS scenario that follows the halogen scenario A1 from WMO (2011). The respective description of RC2 simulation has been modified in the Revised Manuscript (RM) as follows: P4, L3-7 *"More specifically, data from the simulation RC2-base-04 are used, which is part of the set of simulations performed within the ESCiMo project (Jöckel et al., 2016) following the recommendations by the CCMI. According to Eyring et al. (2013), the objective of REF-C2 (RC2) simulations is to*

*produce best estimates of the future ozone and climate changes up to 2100, under specific assumptions about GHG, as well as tropospheric ozone and aerosol precursors that follow RCP 6.0, and a specific ODS scenario that follows the halogen scenario A1 from WMO (2011b)."*

2. Page 4, Lines 23–28. Do the authors take into account any limitations of the work by Škerlak et al. (2015)? How would these affect the comparison between the two methodologies?

For the fold detection we implement the same 3-D labelling algorithm as the one used in the study of Škerlak et al. (2015), thus the methodologies are the same. The differences found compared to Škerlak et al. (2015) are subject to the different meteorological input and the different vertical and horizontal resolution in each case. As Škerlak et al. (2015) use the ERA-Interim dataset, we consider this study as a reference to assess the performance of the RC2 simulation. Given the fact that RC2 is a free-running (without nudging) simulation, the spatiotemporal features of fold frequencies in RC2 are reproduced satisfactorily. Yet, there is an overestimation of fold frequencies compared to Škerlak et al. (2015). The respective discussion has been modified in the RM as follows: P5, L10-16 *"The results are similar, implying a good representation of present-time monthly folding frequency. Yet, a small systematic overestimation of EMAC fold frequencies is seen. Additionally, not only the hemispheric monthly fold frequencies are similar between data from simulation RC2-base-04 and data from ERA-Interim, but also the geographical distribution presents the same patterns (see Fig.4). Any discrepancies might be attributed to the fact that RC2-base-04 is a free-running simulation with different horizontal and vertical resolution. We can therefore consider that the data used in this work are comparable for present-time with state-of-the-art calculations based on the ERA-Interim dataset."*

3. Page 6, Lines 14–23. The strengthening of the BDC would imply more rising air in the tropics, which would then be reflected in a decrease of ozone in the tropical lower stratosphere. Is there any evidence on this, also based on TST (troposphere-to-stratosphere transport) studies? In particular, is Line 19 ("increased upwelling of tropospheric ozone-poor air into the lower stratosphere"), supported by any result? At line 20, the authors indicate a "global STE increase" as the main cause of tropospheric ozone increase, but would this include an increase in both of the two components, i.e., STT and TST, or does it refer to STT only?

The decrease of tropical lower stratospheric ozone under an increase of GHGs due to a BDC strengthening and the induced upwelling enhancement has been reported from several studies, such as Zeng et al. (2010), Young et al. (2013), Banerjee et al. (2016) and Abalos et al. (2017). Specifically, Abalos et al. (2017) suggested an increase in the tropical upwelling, and thus a stronger vertical TST in the future. The decrease of tropical lower stratospheric ozone in EMAC RC2 simulation is presented

in Figure R1.1a, depicting the differences of zonal-mean ozone partial pressure between the FUT and REF periods. Moreover, in Figure R1.1b we present the temperature profiles over the tropics (20S-20N) for the REF and FUT period, as well the difference between them. It seems that the projected warming in the upper troposphere combined with the projected cooling in lower stratosphere results in enhanced upwelling through the tropopause and towards the lower stratosphere, which also agrees with the findings of Lin et al. (2017). The following discussion has been included in the RM: P7, L15-19 *"This tropical lower stratospheric ozone decrease under an increase of GHGs, due to a BDC strengthening and the induced upwelling enhancement, has been reported in other studies as well (e.g. Zeng et al., 2010; Young et al., 2013; Banerjee et al., 2016; Abalos et al., 2017). Specifically, Abalos et al. (2017) using the artificial tracer e90, suggested an increase in the tropical upwelling, and thus a stronger vertical TST in the future."*

Regarding the "global STE increase" we agree with the reviewer, as we indeed refer to "global STT increase". This has been changed accordingly in the RM.

[Figure]

Figure R1.1. a) Zonal-mean O3 partial pressure differences between the FUT and REF periods (colour shaded). Contours depict the zonal-mean O3 partial pressure during the REF period b) Temperature (REF, FUT and differences) profiles over the tropics (20S-20N). The vertical axis stands for pressure (hPa).

4. Page 7, Lines 2–3. In which way is the increase in GHGs concentrations related to the increase in STE of ozone?

Meul et al. (2018) in their sensitivity simulations with EMAC model accounted for GHG increase (RCP8.5) only, ODS decrease only and both, finding that the GHG increase is the main driver of the increased ozone mass flux into the troposphere through the strengthening of the BDC and the increase of the net ozone production in the stratosphere. The respective sentence has been modified in the RM as follows: P7, L30-32 *"Meul et al. (2018) in their future projected simulations under the RCP8.5 GHGs scenario with the EMAC model noted a similar increase in ozone STT through the strengthening of the BDC and the increase of the net ozone production in the stratosphere, which was attributed to the rising GHGs concentrations".*

5. Page 7, Lines 12–14. Again, the role and quantification of TST in not taken into account here. What role would it play in modulating the increase of ozone STE reported in the paper?

An explicit quantification of TST is beyond the scope of this paper, as the EMAC model doesn't include the appropriate tracer (like e90 tracer). However, the effect of TST is shown over the tropics with enhanced upwelling leading to higher water vapour mixing ratios (see Figure R1.2 below) and lower ozone in the lower stratosphere (Figure R1.1a).

[Figure]

Figure R1.2. a) Zonal-mean water vapour mixing ratio a) differences and b) percentage differences between the FUT and REF periods. The vertical axis stands for pressure (hPa).

6. Page 8, Lines 28–31. Would it be possible to "quantify" the effect of these two contributions (i.e., reduction of ozone precursors emissions and increase of ozone STT), so that one could quantitatively see that the ozone decrease due to emissions reduction is effectively canceled out by the global ozone increase due to STT?

The increase of tropospheric ozone due to the STT increase is depicted in Figures 6 and 7. Quantification of the role of ozone precursor's emissions reduction on ozone is not possible since this is not a sensitivity simulation. Nevertheless, to investigate the mechanisms assisting/cancelling the STT-related tropospheric ozone increase, we have calculated the future projected changes of the main ozone chemical production and loss processes, presented in Figure R1.3. Overall, a reduction of net ozone production is projected in the lower and middle troposphere, as a result of a) the reduction of anthropogenic emissions of ozone precursors leading to decrease of ozone production (Prod-HO2) in the lower troposphere and b) the increase of water vapour leading to increase of ozone destruction (Loss-O1D) in the lower and middle troposphere. Moreover, the increase of ozone in the lower troposphere through RO2 probably indicates the impact of the BVOC emissions of ozone precursor's increase due to the global warming. In the upper troposphere, the dominant feature is the increase of ozone production (Prod-HO2) likely resulting from the enhanced lightning NOx emissions, again due to a warmer climate and the associated enhanced convection activity. Both BVOC and lightning NOx emissions in RC2-base-04 simulation are increasing in future (see Figures 3 and 4 in Jöckel et al. (2016)).

[Figure]

Figure R1.3. Differences in zonal-mean O3 production rates (from HO2, methyl peroxy radical and RO2) (top), in zonal-mean O3 loss rates (from HO2, O1D and OH) (middle) and in zonal-mean net O3 production rates (bottom) between the FUT and REF periods. The vertical axis stands for the model levels.

According to the previous, we have modified several parts of the manuscript:

P1, L15-17 *"..due to the decline of ozone precursors emissions and the enhanced ozone loss from higher water vapour abundances, while in the rest of the troposphere ozone shows a remarkable increase owing mainly to the STT strengthening and the stratospheric ozone recovery."*

P6, L31-33 *"This is also the case in the examined simulation, as the projected increase of water vapour mixing ratios is contributing to the decrease of lower tropospheric ozone through its enhanced chemical loss (not shown)."*

P7, L1-3 *"The aforementioned decreases in lower tropospheric ozone, are overcoming the appearing increases in ozone chemical production (not shown), which are likely associated with the enhanced emissions of BVOCs and lightning NOx (see figures 2, 3 and 4 in Jöckel et al. (2016))."*

P7, L7-12 *"These patterns of tropospheric ozone increase are due largely to a global STT increase, linked to stratospheric ozone recovery and a strengthening of BDC, as suggested by previous studies based on simulations with CCMs (Banerjee et al., 2016; Morgenstern et al., 2018). The enhanced lightning NOx, are also likely to act auxiliary in the direction of increasing tropospheric ozone. In the free troposphere, it seems that the beneficial reduction of ozone precursor emissions and the ozone decline due to higher water vapour content, is cancelled out by the projected increase of stratospheric ozone influx and ozone chemical production from BVOC and lighting NOx."*

P8, L7-8 *"..is mostly driven by the strengthening of BDC and the recovery of stratospheric ozone,.."*

P9, L23-25 *"Ozone in the lower troposphere and near the surface decreases under the projected decline in ozone precursor's emissions and the effect of increased water vapour content. In the middle and upper troposphere the projected strengthening of ozone STT contributes to the increase of ozone globally."*

Technical corrections:

1. Page 5, Line 21. "Green contours", please revise Fig. 4 caption, i.e., "black"→"green".

Done.

2. Pag. 6, Lines 28–29. Please check correspondence between Figure numbering and seasons.

Done.

3. Figure 7. "concnentrations"→"concentrations".

Done.

4. Page 7, Line 21. "EM" or "EMME"? Please be consistent.

We thank the Reviewer for the comment. It is EMME. This has been modifies accordingly in the RM.

5. Page 7, Lines 25 and 30. "positevely"!"positively".

Done.

References

Abalos, M., Randel, W. J., Kinnison, D. E., and Garcia, R. R.: Using the Artificial Tracer e90 to Examine Present and Future UTLS Tracer Transport in WACCM, J. Atmos. Sci., 74, 3383–3403, https://doi.org/10.1175/JAS-D-17-0135.1, 2017.

Banerjee, A., Maycock, A. C., Archibald, A. T., Abraham, N. L., Telford, P., Braesicke, P., and Pyle, J. A.: Drivers of changes in stratospheric and tropospheric ozone between year 2000 and 2100, Atmospheric Chemistry and Physics, 16, 2727–2746, https://doi.org/10.5194/acp-16-2727-2016, 2016

Eyring, V., Lamarque, J.-F., Hess, P., et al.: Overview of IGAC/SPARC Chemistry-Climate Model Initiative (CCMI) community simulations in support of upcoming ozone and climate assessments, SPARC Newsletter, 40, 48–66, 2013

Jöckel, P., Tost, H., Pozzer, A., Kunze, M., Kirner, O., Brenninkmeijer, C. A., Brinkop, S., Cai, D. S., Dyroff, C., Eckstein, J., et al.: Earth System Chemistry integrated Modelling (ESCiMo) with the Modular Earth Submodel System (MESSy) version 2.51., Geoscientific Model Development, 9, 2016

Lin, P., D. Paynter, Y. Ming, and V. Ramaswamy, 2017: Changes of the Tropical Tropopause Layer under Global Warming. J. Climate, 30, 1245–1258, https://doi.org/10.1175/JCLI-D-16-0457.1

Škerlak, B., Sprenger, M., Pfahl, 5 S., Tyrlis, E., and Wernli, H.: Tropopause Folds in ERA-Interim: Global Climatology and Relation to Extreme Weather Events, Journal of Geophysical Research: Atmospheres, 2015.

WMO: Scientific Assessment of Ozone Depletion: 2010, Global Ozone Research and Monitoring Project-Report No. 52, 516 pp.,World Meteorol. Organ., Geneva, Switzerland, 2011

Young, P. J., Archibald, A. T., Bowman, K.W., et al.: Pre-industrial to end 21st century projections of tropospheric ozone from the Atmospheric Chemistry and ClimateModel Intercomparison Project (ACCMIP), Atmospheric Chemistry and Physics, 13, 2063–2090, https://doi.org/10.5194/acp-13-  2063-2013, https://www.atmos-chem-phys.net/13/2063/2013/, 2013

Zeng, G., Morgenstern, O., Braesicke, P., and Pyle, J. A.: Impact of stratospheric ozone recovery on tropospheric ozone and its budget, Geophysical Research Letters,37,https://doi.org/10.1029/2010GL042812,https://agupubs.onlinelibrary.wiley.com/doi/abs/10.1029/2010GL042812, 2010.

---

## Author Comment (AC2) · 3 Oct 2019

Note: Reviewer's comments are presented in black font; authors' responses are presented in blue plain font; manuscript text quotations are presented in blue italics font.

Anonymous Referee #2

We would like to thank Reviewer #2 for her/his time devoted and the constructive and helpful comments.

General comment:

The authors explore the roles of future climate change in tropospheric ozone changes using a global chemistry-climate model with artificial stratospheric ozone tracer. The results of this study emphasize the importance of downward transport of stratospheric ozone associated with tropopause folds. You've convinced me that changes in tropopause folds are regulated by upper-level jet. Also, I agree that projected increase of tropospheric ozone is associated with changes in BDC and STT. However, I find the linkage between the presence of folds and changes in ozone is relatively weak. I would expect shallow tropopause folds, which are located above 200hPa, account for the most changes in folding frequency. How do these shallow folding activities affect the ozone near 400-500hPa or even below? We know that summertime large-scale subsidence at 500hPa over Mediterranean is projected to change [Cherchi et al., Clim Dyn (2016)]. Perhaps the large changes in ozone near 400hPa are primarily associated with changes in descent, while the presence of tropopause folds is secondary. Except the one concern I've pointed out, this paper is very well structured and is certainly within the scope of ACP, although improvements can be applied to make it clearer. Therefore, I only have some minor comments.

We thank the Reviewer for the comments, to which we will respond point by point. Indeed the vast majority of tropopause folds are shallow. Nevertheless, considering that the average pressure of the tropopause in the extratropics is about 250 hPa and that the vertical extend ($\Delta p$) of shallow folds range from 50 to 200 hPa below the tropopause, the shallow foldings extend down to approximately 300-450 hPa. Of course, the large scale subsidence over specific regions, such as the summertime EMME, can further transport high ozone concentrations towards lower tropospheric levels in greater timescales. Thus, the folding mechanism enriches the upper and middle troposphere with high ozone concentrations, which might be further vertically transported under favorable meteorological conditions.

1. What's your rationale for choosing RCP6?

Please refer to our response in Reviewers' #1 Specific Comment #1.

2. Ozone is difficult to simulate in models due to biases in photochemistry processes and precursor emissions. Have you evaluated model performance in ozone? Discussion regarding how biases in EMAC would affect the estimated changes is necessary.

All ESCiMo simulations, including the RC2-base-04 simulation, are evaluated in the study by Jöckel et al. (2016) using the BSTCO (Bodeker Scientific combined total column ozone database; Bodeker et al., 2005) for total column ozone, the AURA Microwave Limb Sounder/Ozone Monitoring Instrument (MLS/OMI; Ziemke et al., 2011) for tropospheric and stratospheric partial column ozone and the ozonesonde dataset by Tilmes et al. (2012) for ozone profiles. In general the seasonal cycle and the spatial distribution of total column ozone are well reproduced in the simulation, with an overestimation of up to 9%. The following sentence has been extended in the RM: P4, L17-18 *"A detailed description of the simulation along with a comprehensive evaluation of ozone with satellite and ozonesonde measurements can be found in Jöckel et al. (2016).".*

3. In Fig.1, wintertime medium and deep fold frequency are much higher than those shown in Škerlak et al. (2015). Will it affect your results? Also, it'd be good to address that the climatological distribution of tropopause folds in your model is consistent with what shown in previous studies.

Given the fact that medium and deep folds are very rare to occur, as the order of magnitude of their frequencies in a global scale are -1 and -2 respectively (please mind the x10 and x100 notations in Figure 1) the impact on our results is expected to be very small. As concerns the climatological distribution of tropopause folds in RC2 simulation compared to previous studies, Figure 4 depicts the spatial distribution of tropopause folds frequency (green contours) for the REF period, which is also discussed compared to the climatology of Škerlak et al. (2015) in the manuscript as follows (P6, L8-13): *"The spatial distribution of fold frequencies during the REF period (green contours in Fig. 4), indicates that in principal folds occur in the regions with high zonal wind speed (colour shadings in Fig. 3). Noteworthy are the hotspots over Asia and Middle East during DJF and JJA, and over the southern Indian Ocean during JJA, whereas during the transition seasons the maxima are located over Asia in MAM, and over Asia and southern Indian Ocean in SON, being consistent with the ERA-Interim derived tropopause fold climatology of Škerlak et al. (2015)."*

4. I'm worried whether the future changes of tropopause folds are robust. Have you compared with other models?

To our knowledge, this is the first projection of tropopause fold frequencies under a future scenario, so we are not able to compare with other models and studies.

Detailed comments:

1. P18, Fig.4 caption, "black" -> "green"; "circles" -> "dots"

Done

2. P6, line 13, "contrary" -> "on the contrary"

Done

3. P7, "positevely" -> "positively"

Done

References

Bodeker, G. E., Shiona, H., and Eskes, H.: Indicators of Antarctic ozone depletion, Atmos. Chem. Phys., 5, 2603–2615, doi:10.5194/acp-5-2603-2005, 2005

Jöckel, P., Tost, H., Pozzer, A., Kunze, M., Kirner, O., Brenninkmeijer, C. A., Brinkop, S., Cai, D. S., Dyroff, C., Eckstein, J., et al.: Earth System Chemistry integrated Modelling (ESCiMo) with the Modular Earth Submodel System (MESSy) version 2.51., Geoscientific Model Development, 9, 2016

Škerlak, B., Sprenger, M., Pfahl, 5 S., Tyrlis, E., and Wernli, H.: Tropopause Folds in ERA-Interim: Global Climatology and Relation to Extreme Weather Events, Journal of Geophysical Research: Atmospheres, 2015.

Tilmes, S., Lamarque, J.-F., Emmons, L. K., Conley, A., Schultz, M. G., Saunois, M., Thouret, V., Thompson, A. M., Oltmans, S. J., Johnson, B., and Tarasick, D.: Technical Note: Ozonesonde climatology between 1995 and 2011: description, evaluation and applications, Atmos. Chem. Phys., 12, 7475-7497, doi:10.5194/acp-12-7475-2012, 2012.

Ziemke, J. R., Chandra, S., Labow, G. J., Bhartia, P. K., Froidevaux, L., and Witte, J. C.: A global climatology of tropospheric and stratospheric ozone derived from Aura OMI and MLS measurements, Atmos. Chem. Phys., 11, 9237–9251, doi:10.5194/acp-11-9237-2011, 2011

---

## Author Comment (AC3) · 3 Oct 2019

Note: Reviewer's comments are presented in black font; authors' responses are presented in blue plain font; manuscript text quotations are presented in blue italics font.

Anonymous Referee #3

We would like to thank Reviewer #3 for her/his time devoted and the constructive and helpful comments.

General comment:

This is an interesting and well-presented paper. My main concern is that like many of the past papers that discuss the influence of climate change (either past or future) on tropospheric ozone, it is difficult to separate out the individual effects of different processes. You appear to be assuming that all (or the vast majority) of your signal is just the combination of changes in STT, together with changes of anthropogenic emissions under RCP6.0. But what about changes in water vapour, and natural emissions from lightning NOx and BVOCs (etc.)? Most of these are barely discussed in the paper, but I think they must be simultaneously changing, and having potentially large effects. Some authors have attempted to separate out some of these processes in the past (e.g., Wild, 2007; Doherty et al.,2013), but this is not easy. This wider context needs to be discussed to place some perspective on where changes in the STT rank compared to other climate change effects on ozone. If this can be included, and the points below, then I am happy to recommend publication in ACP.

We thank the Reviewer for the comments, to which we will respond point by point. We agree with the main comment of the Reviewer, regarding the future projected role of water vapour, lightning NOx and BVOCs on tropospheric ozone changes, and thus, we have included the appropriate discussion in the RM (Introduction, Methodology and Results). In the examined simulation, lighting NOx, soil NOx and BVOC emissions are online calculated by the MESSy submodels LNOX (Tost et al., 2007) and ONEMIS (Kerkweg et al., 2006), and therefore they consider the climate change and the induced effects on ozone chemical production/loss.

Specific comments

P1

L4: Clarify the temporal and spatial context of the 3% increase (i.e. from (1970-99) to (2070-99); is it a global average number, or related to the model grid size?)

The exceedance of 3% increase in fold frequency is seen over some regions. We have modified the respective phrase in the Revised Manuscript (RM) as follows: P1, L4-5 *"Statistically significant changes in tropopause fold frequencies from 1970-99 to 2070-99 are identified in both Hemispheres, regionally exceeding 3%,.."*.

L8: maxima -> largest

Done

L9: Highest background fold frequencies, or changes?

It is the "highest fold frequencies changes". This has been modified in the RM.

Abstract: How is the (likely) shortened lifetime of tropospheric ozone in future, due to higher levels of water vapour, and hence bigger flux through O(1D)+H2O, taken into account? Also what about changes in lightning NOx emissions (and BVOC emissions, and other climate dependent processes…) that may affect tropospheric O3? Introduction: This should also mention other climate-driven influences on tropospheric O3 – ie water vapour, lightning NOx, biogenic VOC emissions, etc.

We agree with the Reviewer and thus we have included the following discussion in the RM:

Introduction, P2, L6-9 *"Moreover, climate-related changes in lightning NOx emissions, Biogenic Volatile Organic Compounds (BVOCs) emissions and water vapour content, are also key drivers of future tropospheric ozone changes, affecting its chemical production and loss processes (Wild, 2007; Fiore et al., 2012, 2015; Doherty et al., 2013)".*

Introduction, P3, L9-19: *"There is a high confidence that the increasing temperature will lead in a decline of lower tropospheric ozone through the enhanced water vapour abundances and the associated acceleration of ozone chemical loss (Fiore et al., 2012, 2015; Fu and Tian, 2019). Several studies indicate that the emissions of BVOCs are subject to increase in a warming climate, as they are temperature-sensitive, leading to a positive feedback on future ozone chemical production (Zeng et al., 2008; Weaver et al., 2009; Doherty et al., 2013). Yet, other studies considering the CO2 inhibition effect, report that this positive feedback on ozone may be offseted or even reverse negative (Tai et al., 2013; Hantson et al., 2017). Climate-related changes in lightning activity and the associated NOx emissions are thought to have complex implications for tropospheric ozone. While the enhancement in lightning NOx emissions in a warmer climate will increase baseline ozone, the induced enhancement in OH will result in CH4 reduction and thus, in a decline of ozone chemical production on greater timescales (Wild, 2007; Banerjee et al., 2014; Murray, 2016). Moreover, climate-induced changes in NOx emissions from soils and ozone precursors emissions from wildfires are also expected to modulate future ozone changes (Voulgarakis and Field, 2015; Romer et al., 2018)."*

Methodology, P4, L15-17: *"Lightning NOx emissions and emissions of BVOCs are online calculated by the MESSy submodels LNOX (Tost et al., 2007) and ONEMIS (Kerkweg et al., 2006), respectively, considering the effects of climate change."*
P4
L1: Presumably the stratospheric ozone tracer ignores rapid cycling processes involving O3, ie.: O3 + NO -> NO2 + O2 NO2 + hv -> NO + O O2 + O -> O3 Which

form a null cycle. But presumably it does include Ox (O3+NO2) loss processes that interact with this cycle, such as O(1D)+H2O and NO2 dry deposition?

Yes, this is correct.

P5
L7 in the Northern Hemisphere, not at the Northern Hemisphere (and several other similar instances). 'At' is appropriate for a specific site, whereas 'in' is more appropriate for a larger region. I don't think this is just my dubious grammar.

Done

L22 Do you mean the hotspots in the REF distribution, or the changes?

We mean for the REF period. This has been modified in the RM to make it clearer (P6, L10).

L26 delete 'a'

Done

L27 It is a bit confusing that Figure 3 has colours for REF winds and contours for FUTREF changes, whilst Figure 4 has contours for REF fold frequencies and colours for changes. I suggest all the figures follow a consistent format?

We agree with the comment. Figure 3 is in the same format as Figure 4 in the RM. Figure 3 caption has been modified accordingly also.

P6
L12 lower tropospheric ozone?

We thank the Reviewer for the comment. We mean that "Clearly, temperature and humidity under a warmer climate play an important role in decreasing tropospheric ozone in the tropical Pacific, due to the increased rate of the ozone destruction reactions (Revell et al., 2015)", which has been updated in the RM (P6L33-P7L1). Moreover, we have updated the reference of Revell et al. (2015) with the appropriate one (P14, L28-30), as initially we have inadvertently included another one.

Section 3.2 What about lightning NOx? Does it change? And BVOCs?

Please, also see our response to Reviewer's #1 Specific Comment #6, where we present the future changes in ozone chemical production and loss. The future projections of soil NOx, total BVOCs, and lightning NOx emissions for the examined simulation (RC2-base-04) are provided in Figures 2, 3 and 4, respectively, in Jöckel et al., (2016), depicting an increase up to 2100. As the examined simulation is not sensitivity, we cannot separate the respective effects on ozone. Nevertheless, we have included the following discussion regarding the potential effects of both in future ozone changes.

P1, L15-17 *"..due to the decline of ozone precursors emissions and the enhanced ozone loss from higher water vapour abundances, while in the rest of the*

*troposphere ozone shows a remarkable increase owing mainly to the STT strengthening and the stratospheric ozone recovery."*

P6, L31-33 *"This is also the case in the examined simulation, as the projected increase of water vapour mixing ratios is contributing to the decrease of lower tropospheric ozone through its enhanced chemical loss (not shown)."*

P7, L1-3 *"The aforementioned decreases in lower tropospheric ozone, are overcoming the appearing increases in ozone chemical production (not shown), which are likely associated with the enhanced emissions of BVOCs and lightning NOx (see figures 2, 3 and 4 in Jöckel et al. (2016))."*

P7, L7-12 *"These patterns of tropospheric ozone increase are due largely to a global STT increase, linked to stratospheric ozone recovery and a strengthening of BDC, as suggested by previous studies based on simulations with CCMs (Banerjee et al., 2016; Morgenstern et al., 2018). The enhanced lightning NOx, are also likely to act auxiliary in the direction of increasing tropospheric ozone. In the free troposphere, it seems that the beneficial reduction of ozone precursor emissions and the ozone decline due to higher water vapour content, is cancelled out by the projected increase of stratospheric ozone influx and ozone chemical production from BVOC and lighting NOx."*

P8, L7-8 *"..is mostly driven by the strengthening of BDC and the recovery of stratospheric ozone,.."*

P9, L23-25 *"Ozone in the lower troposphere and near the surface decreases under the projected decline in ozone precursor's emissions and the effect of increased water vapour content. In the middle and upper troposphere the projected strengthening of ozone STT contributes to the increase of ozone globally."*

Section 3.3 From your experiments it is not possible to separate the effects of stratospheric O3 recovery (due to ODS declines) and enhanced STE. Is that correct?

Yes, this is correct.

P7
L25 and l30 positively

Done

Section 4: Should EM be EMME?

Yes it should. Every instance of EM is replaced by EMME.

P18, Figure 4 caption: green not black. What are the units of fold frequency? "hatched with black circles" -> "indicated by black dots".

Done. The units of fold frequency are percentage (%) of fold occurrence during the respective period.

References

[revised manuscript text omitted]